# Influence of Fly Ash on Mechanical Properties and Hydration of Calcium Sulfoaluminate-Activated Supersulfated Cement

**DOI:** 10.3390/ma13112514

**Published:** 2020-05-31

**Authors:** Zhengning Sun, Jian Zhou, Qiulin Qi, Hui Li, Na Zhang, Ru Mu

**Affiliations:** School of Civil and Transportation Engineering, Hebei University of Technology, Tianjin 300401, China; zhengning0211@163.com (Z.S.); qiulin183@126.com (Q.Q.); hla_zyj@hebut.edu.cn (H.L.); zn2011@126.com (N.Z.); ru_mu@hotmail.com (R.M.)

**Keywords:** supersulfated cement, calcium sulfoaluminate, hydration, fly ash, ettringite

## Abstract

This paper aimed to report the effects of fly ash (FA) on the mechanical properties and hydration of calcium sulfoaluminate-activated supersulfated cement (CSA-SSC). The CSA-SSC comprises of 80% granulated blast furnace slag (GBFS), 15% anhydrite, and 5% high-belite calcium sulfoaluminate cement (HB-CSA) clinker. The hydration products of CSA-SSC with or without FA were investigated by X-ray diffraction and thermogravimetric analysis. The experimental results indicated that the addition of FA by 10% to 30% resulted in a decrease in the rate of heat evolution and total heat evolution of CSA-SSC. As the content of FA was increased in the CSA-SSC system, the compressive and flexural strengths of the CSA-SSC with FA after 1 day of hydration were decreased. After 7 days of hydration, the compressive and flexural strength of CSA-SSC mixed with 10 wt.% and 20 wt.% of FA rapidly increased and exceeded that of ordinary Portland cement (OPC), especially the flexural strength. Moreover, the compressive strength of CSA-SSC mixed with 30 wt.% of FA after 90 days of hydration was close to that of OPC, and flexural strength of CSA-SSC mixed with 30 wt.% of FA after 7 days of hydration was close to that of OPC. The hydration products of the CSA-SSC and CSA-SSC mixed with FA were mainly ettringite and calcium silicate hydrate (C-S-H).

## 1. Introduction

Ordinary Portland cement (OPC) is the most widely used cementitious material in civil engineering construction. The hydration reaction of OPC paste is accompanied by the release of a vast amount of hydration heat, which enhances the internal temperature of the mass concrete structure. When the thermal deformation is restrained, it is easy to cause cracking of the concrete surface and, thus, seriously affect the durability and safety of the mass concrete structure. Therefore, weakening the temperature rising inside the mass concrete structure is a critical key to prevent the development of cracks from the mass concrete structure [1,2,3]. The appropriate selection of cementitious material type is considered to be a simple and effective method to control the formation of cracks. The cementitious material with low heat evolution should be widely used in mass concretes [4,5,6].

Supersulfated cement (SSC) has the characteristics of low hydration heat and the trace amounts of clinker, which is an environmentally friendly cement. SSC is comprised of 80%−85% granulated blast furnace slag (GBFS), 10%−15% gypsum or anhydrite as the sulfate activator, and <5% ordinary Portland cement (OPC) clinker or lime as the alkali activator [7,8,9,10]. GBFS, with a high Al_2_O_3_ content (above 13%), has been found to be suitable for the manufacture of SSC, and it has exhibited a reasonable strength development [11]. Anhydrite or gypsum can also be replaced by other industrial by-products, such as phosphogypsum and fluorgypsum [12,13,14,15]. Besides, alkalis, such as NaOH and KOH, are utilized in SSCs as alkaline activators. The SSC requires less energy and produces lower greenhouse gas emissions than the OPC [16]. In addition to the environmental advantages, the SSC exhibits chemical or thermal behaviors, such as high durability in sulfate solutions, low hydration heat, and excellent resistance to chloride, acid, and seawater attacks, which all confirm that the SSC is an alternative to Portland cement [17,18,19,20,21,22].

In 1908, Hans Kühl, a German engineer, first discovered that sulfates could be used as an activator to activate the activation of GBFS, opening up works of utilizing GBFS to produce SSC [9]. Since the 1920s, SSC was first produced and used in France and Belgium. From 1940 to 1965, due to a shortage of Portland cement clinker, SSC was increasingly used in European countries. In the 1970s, as the German iron manufacturing process was changed, the Al_2_O_3_ content in GBFS was reduced. Therefore, GBFS with the low Al_2_O_3_ content became less reactive and had a negative effect on the development of the strength of SSC. This led to the disappearance of SSC from the market and the withdrawal of the German standards about SSC.

The hydration products of SSC are mainly ettringite and calcium silicate hydrate (C-S-H), as well as a small amount of monosulfoaluminate (AFm) and OH-hydrotalcite (Mg_4_Al_2_(OH)_14_·3H_2_O) like gel [23,24,25,26]. The formation mechanism of ettringite mainly affects the early strength of SSC, and the ettringite is formed by reacting to the Al_2_O_3_ in the GBFS with anhydrate. Therefore, the early strength of SSC mainly depends on the content of Al_2_O_3_ in GBFS. The late strength of SSC mainly depends on the amount of C-S-H produced [26].

Calcium sulfoaluminate-activated supersulfated cement (CSA-SSC) is a novel type of cementitious material with 80 wt.% GBFS as the main raw material that is activated by 15 wt.% anhydrite and 5 wt.% high-belite calcium sulfoaluminate cement (HB-CSA) clinker. The clinker contains 37–47% belite (C_2_S), 20–35% of ye’elimite (C_4_A_3_$\bar{S}$), 14–26.3% calcium sulfate (CaSO_4_), and 0.5% to 4.6% of free calcium oxide (f-CaO) [27]. CSA-SSC has excellent mechanical properties and a low hydration heat. The GBFS in the cement does not need to be ultra-fine ground and has a specific surface area of 4000 cm^2^/g. It can be produced by mixing GBFS powder with activator or co-grinding process, and other mineral admixtures and industrial by-products can also be added to the CSA-SSC system. Therefore, CSA-SSC has extremely low CO_2_ emissions caused by raw materials, and its energy consumption is greatly reduced. In order to further reduce the hydration heat of CSA-SSC, fly ash (FA) is added into the CSA-SSC system.

The CSA-SSC system is mixed with FA from 10 wt.% to 30 wt.% to reduce the hydration heat and improve the utilization rate of industrial by-products. In this study, the mechanical properties of CSA-SSC mixed with FA were compared and analyzed with the strength of reference mortar prepared by CSA-SSC without FA and OPC. The rate of heat evolution and cumulative heat of CSA-SSC, CSA-SSC mixed with FA, and OPC were measured with isothermal conduction calorimetry (ICC). The hydration process of CSA-SSC and CSA-SSC mixed with FA was studied by X-ray diffraction (XRD) and thermogravimetric analysis (TGA).

## 2. Materials and Methods

### 2.1. Materials

The CSA-SSC consisted of 80 wt.% GBFS, 15 wt.% anhydrite, and 5 wt.% HB-CSA clinker. HB-CSA clinker mainly consisted of C_4_A_3_$\bar{S}$, C_2_S, f-CaO, and CaSO_4_ [27]. The mixes were designed with four levels of CSA-SSC replaced by 0%, 10%, 20%, and 30% (by weight) of FA. The chemical compositions of CSA-SSC, FA, and OPC were determined by titrimetry, given in Table 1. The mineralogical composition of the HB-CSA clinker is described in Table 2.

### 2.2. Methods

#### 2.2.1. Mechanical Properties

The compressive and flexural strengths of CSA-SSC, 90% CSA-SSC + 10% FA, 80% CSA-SSC + 20% FA, 70% CSA-SSC + 30% FA, and OPC mortars at 1, 3, 7, 14, 28, and 90 days were measured according to the European Standard DIN EN 196-1 [28]. The mixture proportions of the mortars are given in Table 3. The mortars were cast into 40 mm × 40 mm × 160 mm prisms. The samples were placed in a moist cabinet at a temperature of 20 °C and relative humidity of 95% for 24 h. After demolding, the samples were placed in water (20 ± 1) for curing. Six specimens were tested for compressive strength, and three specimens were tested for flexural strength. The compressive and flexural strengths were determined by averaging the test results.

#### 2.2.2. Isothermal Conduction Calorimetry

The heat evolution of CSA-SSC, 90% CSA-SSC + 10% FA, 80% CSA-SSC + 20% FA, 70% CSA-SSC + 30% FA, and OPC paste was investigated with an ICC (thermometric TAM air isothermal calorimeter) (TA Instruments, New Castle, DE, USA) at 20 °C within 168 h. The sample (2.0 ± 0.05 g) was used, and the water-to-cement ratio was 0.4. The mixture was stirred for 1 min by internal stirring. The hydration of heat of the paste was immediately recorded after water was added.

#### 2.2.3. Sample Preparation for X-ray Diffraction (XRD) and Thermogravimetric Analysis (TGA)

The paste samples were used for XRD and TGA analysis to study the hydration products of the CSA-SSC and 70% CSA-SSC + 30% FA at 1, 7, 14, 28, and 90 days. The mixture proportions of the pastes are given in Table 4. The CSA-SSC and 70% CSA-SSC + 30% FA pastes were casted into 40 mm × 40 mm × 160 mm prisms. There was one test piece for each age specimen. The specimens were placed in a moist cabinet at a temperature of 20 °C and relative humidity of 95% for 24 h. After demolding, the specimens were placed in water (20 ± 1 °C) for curing.

At the predetermined curing ages, the prisms of the CSA-SSC and 70% CSA-SSC + 30% FA paste were crushed into small particles (about 1 g). Solvent exchange with ethanol was used to remove water from the particle samples for terminating cement hydration. The crushed pastes were placed in ethanol for 4 h and then moved into another ethanol for 24 ± 1 h. The crushed pastes were placed in a 40 °C drying cabinet and dried for 24 ± 1 h. A part of the dried samples was ground into fine powder for XRD and TGA experimental tests by passing through a sieve (60 µm).

#### 2.2.4. X-Ray Diffraction

The XRD analysis was used to identify the hydration products of the specimens at 1, 3, 7, 28, and 90 d. A Bruker D8 Advance diffractometer with CuKα radiation was used (Bruker, Karlsruhe, Germany). The CuKα X-rays were generated using 40 mA and 40 kV tube operating conditions. All scans were measured over a 2θ angular range from 5–55°, using a step size of 0.02° with a dwell time of 0.3 s. The single-crystal structures of the crystalline phases were taken from the Inorganic Crystal Structure Database (ICSD, 2006) [29].

#### 2.2.5. Thermogravimetric Analysis

The TGA analysis was to confirm the amorphous phases, such as C-S-H and OH-hydrotalcite, and identify the hydration products previously explored by XRD analysis. The TGA analysis was performed using a Mettler Toledo TGA/DSC1 (TA Instruments, Mettler Toledo, Columbus, OH, USA). The TGA analysis was conducted in 30-μL aluminum oxide vessels filled with 20 ± 0.2 mg hardened powder pastes at 1, 3, 7, 28, and 90 days of curing age. The dynamic heating ramp ranged from 30 to 600 °C with a heating rate of 10 °C/min. The test was conducted under an N_2_ atmosphere. The TGA curves of some hydration products in the samples overlapped, such as gypsum, C-S-H, ettringite, and OH-hydrotalcite. The TGA results of each hydration product were used to analyze the hydrated cement samples. The temperature range and weight loss peaks for major hydration products, identified in differential thermogravimetry (DTG) curves, are given in Table 5.

## 3. Results

### 3.1. Strength Development

The compressive and flexural strengths of CSA-SSC, CSA-SSC with FA, and OPC mortars after 1, 3, 7, 14, 28, and 90 days hydration are plotted in Figure 1 and Figure 2, respectively. After 1 day of hydration, the compressive strength of CSA-SSC mortar was higher than that of the 90% CSA-SSC + 10% FA, 80% CSA-SSC + 20% FA, and 70% CSA-SSC + 30% FA mortar and slightly lower than that of OPC mortar. After 3 days of hydration, the compressive strength of CSA-SSC mortar rapidly increased and began to be higher than that of OPC mortar. After 28 days of hydration, the compressive and flexural strengths of CSA-SSC mortar were 68.0 MPa and 12.3 MPa, respectively. The CSA-SSC exhibited higher compressive strengths than that of 90% CSA-SSC + 10% FA, 80% CSA-SSC + 20% FA, and 70% CSA-SSC + 30% FA at all hydration ages. The CSA-SSC exhibited higher late compressive strengths and a larger increase in the compressive strength than the OPC.

As the content of FA was increased in the CSA-SSC system, the compressive and flexural strengths of CSA-SSC with FA mortar were decreased. From 1–14 days, the flexural strength of the CSA-SSC, 90% CSA-SSC + 10% FA, 80% CSA-SSC + 20% FA, and 70% CSA-SSC + 30% FA mortar rapidly increased. However, the flexural strength of the 90% CSA-SSC + 10% FA and 80% CSA-SSC + 20% FA mortar after 7 days of hydration was higher than that of OPC. The compressive strength of the 70% CSA-SSC + 30% FA after 90 days of hydration was close to that of OPC. Moreover, after 7 days of hydration, the flexural strength of the 70% CSA-SSC + 30% FA was close to that of OPC. The strength of CSA-SSC, 90% CSA-SSC + 10% FA, and 80% CSA-SSC+20% FA mortar fulfilled the requirements specified in the European standard EN 15743:2015 for the SSC of strength class 52.5 N, which states the compressive strengths at 28 days should not be lower than 52.5 MPa, respectively [31].

### 3.2. Heat Evolution 

The rate of heat evolution of the CSA-SSC, CSA-SSC with FA, and OPC pastes, measured using ICC at a period of 168 h, is plotted in Figure 3, respectively. At the early stage of hydration (0–24 h), the CSA-SSC, CSA-SSC with FA, and OPC pastes showed two peaks in the curves of the rate of heat evolution. The first peak was extremely intense and of a short duration, which probably was due to the wetting and dissolution of cement powders. The duration of the initial reaction of the CSA-SSC, 90% CSA-SSC + 10% FA, 80% CSA-SSC + 20% FA, and 70% CSA-SSC + 30% FA pastes was 0.5 h (0–0.5 h), whereas that of the OPC paste was 1 h (0–1 h). As the content of FA was increased in the CSA-SSC system, the rate of heat evolution of CSA-SSC with FA paste was accelerated during 1–4 h of hydration, but the duration of the induction period of CSA-SSC with FA paste was shortened.

The second major peak was considered to be the main hydration process of the cement components. The heat evolution rate of the CSA-SSC paste was characterized by the narrow main hydration peak and the highest peak intensity with a maximum at ~7 h. The acceleration period of CSA-SSC paste lasted for ~2 h (5–7 h). HB-CSA clinker contained C_4_A_3_$\bar{S}$, which had a fast hydration rate, a concentrated exotherm, and a short duration. Therefore, the acceleration period of CSA-SSC, 90% CSA-SSC + 10% FA, 80% CSA-SSC + 20% FA, and 70% CSA-SSC + 30% FA pastes was shorter compared with that of the OPC paste. As the content of FA was increased in the CSA-SSC system, the second major peak of CSA-SSC with FA paste was obviously decreased. The highest peak intensity with a maximum of OPC paste occurred at ~10 h, and the acceleration period lasted for ~9 h (8–17 h). Compared with the second major peak of CSA-SSC and CSA-SSC mixed with FA pastes, the second major peak of OPC paste appeared later, and the duration of the acceleration period was longer.

During the first 6 h of hydration, the cumulative heat of CSA-SSC, 90% CSA-SSC + 10% FA, 80% CSA-SSC + 20% FA, 70% CSA-SSC + 30% FA, and OPC pastes slowly increased, as shown in Figure 4. During the period of 8–32 h, the cumulative heat of OPC paste increased rapidly, whereas that of CSA-SSC and CSA-SSC with FA pasted exhibited a relatively gentle curve. After 96 h hydration, both the cumulative heat curve of CSA-SSC, 90% CSA-SSC + 10% FA, 80% CSA-SSC + 20% FA, 70% CSA-SSC + 30% FA, and OPC pastes tended to be horizontal. The total heat of CSA-SSC paste was only 191 J/g at 168 h. As the content of FA was increased in the CSA-SSC system, the total heat of CSA-SSC mixed with FA was further decreased.

### 3.3. Hydration Products

#### 3.3.1. XRD

The XRD patterns of the CSA-SSC and 70% CSA-SSC + 30% FA paste after 1, 3, 7, 14, 28, and 90 days hydration are plotted in Figure 5. The XRD results indicated that ettringite was the main crystalline hydration product during the hydration of the CSA-SSC and 70% CSA-SSC + 30% FA paste. After 1 day of hydration, the C_4_A_3_$\bar{S}$ in the CSA-SSC and 70% CSA-SSC + 30% FA paste was not observed. This indicated that the C_4_A_3_$\bar{S}$ was completely hydrated at an early age. With the increase in age, as the anhydrite and gypsum in the CSA-SSC and 70% CSA-SSC + 30% FA paste were gradually consumed, the amount of ettringite was gradually increasing. The hydration products of the 70% CSA-SSC + 30% FA paste were similar to those of CSA-SSC paste, but the amount of hydration products formed was different. By using the intensity of diffraction peak as a reference, the diffraction peak of ettringite formed by the 70% CSA-SSC + 30% FA paste was lower than that formed by CSA-SSC paste. Moreover, after 1 day of hydration, a large amount of anhydrite in the 70% CSA-SSC + 30% FA paste was converted into gypsum, and the gypsum was involved in the hydration reaction. During the hydration process of the CSA-SSC and 70% CSA-SSC + 30% FA pastes, a small amount of C_2_S participated in the reaction of the GBFS, but its reaction was slow. The Ca(OH)_2_, C_2_S, and AH_3_ phases were not found in the CSA-SSC and 70% CSA-SSC + 30% FA pastes at each hydration age. The Ca(OH)_2_ produced by the hydration of f-CaO and C_2_S might be consumed in the reaction of the GBFS. The amorphous phase, which could not be elucidated using XRD, mainly consisted of the GBFS, C-S-H gel, and OH-hydrotalcite gel, which were characterized using TGA.

#### 3.3.2. TGA

The TGA and DTG curves of the CSA-SSC and 70% CSA-SSC + 30% FA pastes after at 1, 3, 7, 14, 28, and 90 days hydration are shown in Figure 6 and Figure 7, respectively. The DTG data of the CSA-SSC and 70% CSA-SSC + 30% FA pastes had overlapping signals that occurred for the hydration phases C-S-H, ettringite, and gypsum. Therefore, the hydration products of the CSA-SSC and 70% CSA-SSC + 30% FA pastes were qualitatively analyzed by using different pure phases as reference. The data of pure phases are given in Table 2. The hydration products of the CSA-SSC and 70% CSA-SSC + 30% FA pastes were mainly ettringite and C-S-H; however, small amounts of gypsum and OH-hydrotalcite were also present. The Ca(OH)_2_, AFm, and Al(OH)_3_ gels were not found. After 1 day of hydration, a considerable amount of ettringite was formed in the CSA-SSC and 70% CSA-SSC + 30% FA pastes, followed by C-S-H and a small amount of gypsum. After 3 days of hydration, the weight loss peak of gypsum disappeared in the CSA-SSC and 70% CSA-SSC + 30% FA paste. At first, after 28 days of hydration, the quantity of C-S-H produced increased. After 90 days of hydration, there was a significant weight loss peak of the C-S-H in CSA-SSC paste, thereby indicating that a large amount of C-S-H was formed in the late hydration stage. However, from 28 to 90 days of hydration, the weight loss peak of ettringite in CSA-SSC paste slightly increased. The trend indicated that the formation of ettringite in the CSA-SSC paste slowly increased in the late stage of hydration.

From 3 to 90 days of hydration, the amount of C-S-H and ettringite produced by the 70% CSA-SSC + 30% FA paste continued to increase. In addition, both CSA-SSC and 70% CSA-SSC + 30% FA pastes contained a small amount of OH-hydrotalcite, which slightly increased with the hydration processing. The weight loss peak of the 70% CSA-SSC + 30% FA paste was less than that of CSA-SSC paste at each hydration age, indicating that the amount of hydration products formed by the 70% CSA-SSC + 30% FA paste was less than that of CSA-SSC paste.

## 4. Discussion

The HB-CSA clinker contents are highly reactive minerals—C_4_A_3_$\bar{S}$, f-CaO, and CaSO_4_. The CSA-SSC matrix shows high reactivity and increased early and late strengths. The reaction of f-CaO and C_2_S in HB-CSA clinker provides an alkaline environment for the activation of GBFS and FA, i.e., to break the Si–O–Si, Si–O–Al, and Al–O–Al bonds on the surface of GBFS and FA and to promote its dissolution [26]. In the low alkaline environment, the active minerals on the surface of GBFS firstly dissolved, as well as the dissolved active minerals containing CaO and Al_2_O_3_ reacted with anhydrite to form ettringite. After 1 day of hydration, the formation of ettringite was observed, which constituted the initial framework of the matrix to provide the early strength of CSA-SSC and CSA-SSC with the FA matrix. The early hydration products of CSA-SSC and CSA-SSC with FA pastes were mainly ettringite. After the depletion of anhydrite, the reaction of GBFS with H_2_O generated C-S-H, resulting in the continuous strength increase.

The glass phase on the surface of FA was relatively thicker, so it was necessary to consume more OH^−^ to dissolve its glass phase. In the low alkali environment, the dissolution rate of FA in the CSA-SSC with the FA matrix was slower than that of GBFS. In the early stage of hydration, FA mainly acted as particle filling in the CSA-SSC matrix, which had little contribution to the early strength of CSA-SSC with the matrix. Moreover, the higher the content of FA, the lower the early strength of the CSA-SSC matrix. However, after 7 days of hydration, the compressive and flexural strengths of CSA-SSC mixed with 10 wt.% and 20 wt.% of FA were higher than that of OPC. After 90 days of hydration, the compressive and flexural strengths of CSA-SSC mixed with 30 wt.% of FA were higher than that of OPC, which indicated that the pozzolanic effect of FA improved the strength. Therefore, the addition of FA to the CSA-SSC matrix made a contribution to late strengths.

As the content of FA was increased in the CSA-SSC system, the second major peak of CSA-SSC with FA paste was obviously decreased and appeared earlier, and the cumulative heat of CSA-SSC with FA paste was also significantly decreased. The reason for this phenomenon might be due to the water-reducing effect of glassy FA, which caused an increase in the effective water-cement ratio of the CSA-SSC with FA and led to the rapid hydration of CSA-SSC with FA at an early stage of hydration.

The relationship between the chemically bonded water obtained by TGA and the compressive strength of the CSA-SSC and 70% CSA-SSC + 30% FA is shown in Figure 8. In general, the amount of chemically bonded water of hydrated paste was proportional to the amount of hydration products and to the degree of hydration. The chemically bonded water of the CSA-SSC and 70% CSA-SSC + 30% FA was linear with the compressive strength of the matrix. The compressive strength of CSA-SSC was significantly higher than that of the 70% CSA-SSC + 30% FA at the same age. Moreover, the amount of hydration products of the CSA-SSC was more than that of the 70% CSA-SSC + 30% FA at the same hydration age. The high compressive strength of CSA-SSC might be due to a large amount of formation of hydration products.

## 5. Conclusions

Based on the investigation of the influence of FA on hydration heat and mechanical properties of CSA-SSC, the following conclusions were drawn:The addition of FA to CSA-SSC resulted in a decrease in compressive and flexural strengths proportionally.The CSA-SSC mixed with FA did not change the duration of the initial reaction but shortened the duration of the induction period. The acceleration period of the CSA-SSC and CSA-SSC with FA pastes was shorter compared with that of the OPC paste. The cumulative heat of the CSA-SSC was only 191 J/g after 168 h of hydration. As the content of FA was increased in the CSA-SSC system, the cumulative heat of the CSA-SSC with FA was further reduced.The CSA-SSC and CSA-SSC with FA mainly formed ettringite in the early stage, and C-S-H was formed in the late stage. The amount of hydration products of the CSA-SSC was more than that of the CSA-SSC with FA.The compressive strength of the CSA-SSC and 70% CSA-SSC + 30% FA had a linear growth relationship with the amount of chemically bound water.

## Figures and Tables

**Figure 1 materials-13-02514-f001:**
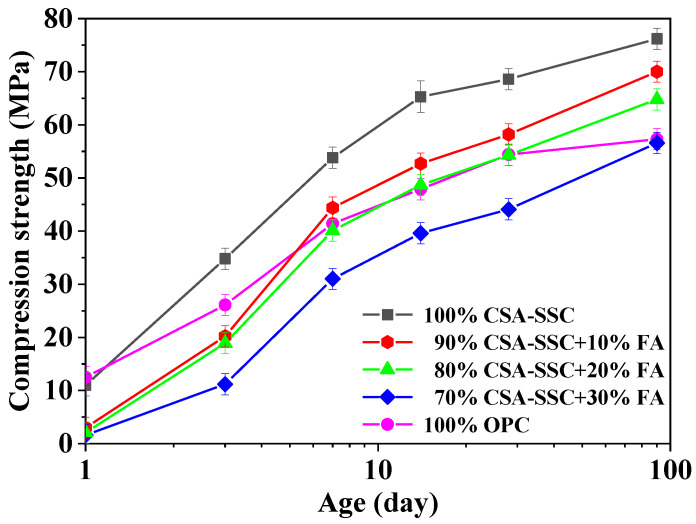
Compressive strength development of the calcium sulfoaluminate-activated supersulfated cement (CSA-SSC), CSA-SSC with fly ash (FA), and ordinary Portland cement (OPC) mortars.

**Figure 2 materials-13-02514-f002:**
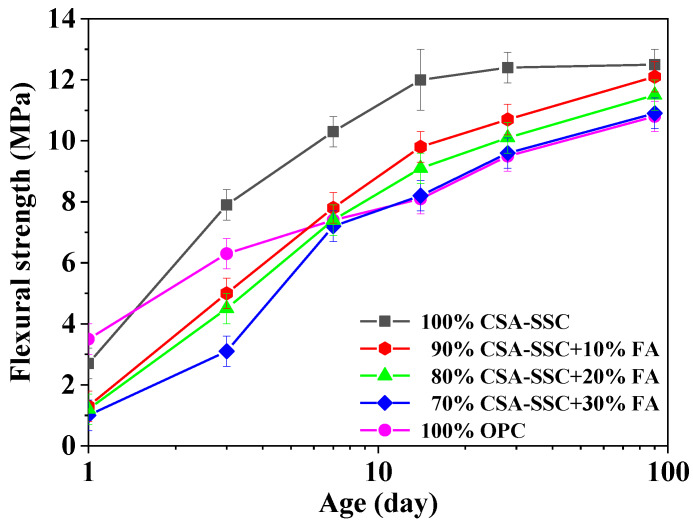
Flexural strength development of the CSA-SSC, CSA-SSC with FA, and OPC mortars.

**Figure 3 materials-13-02514-f003:**
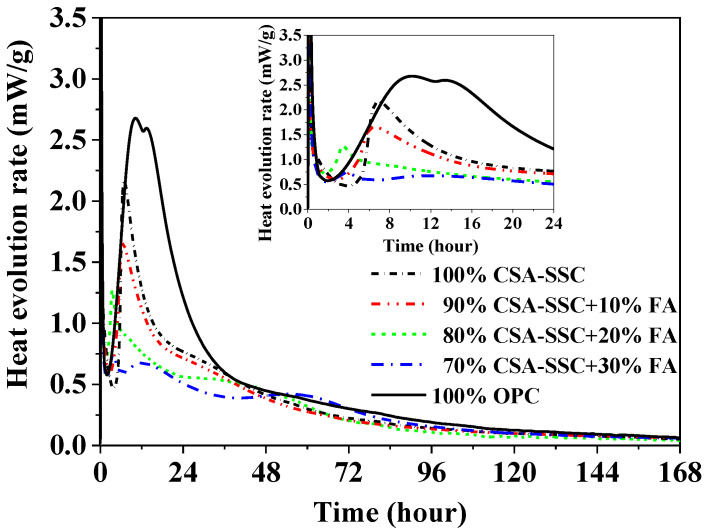
Rate of heat evolution of CSA-SSC, CSA-SSC with FA, and OPC pastes.

**Figure 4 materials-13-02514-f004:**
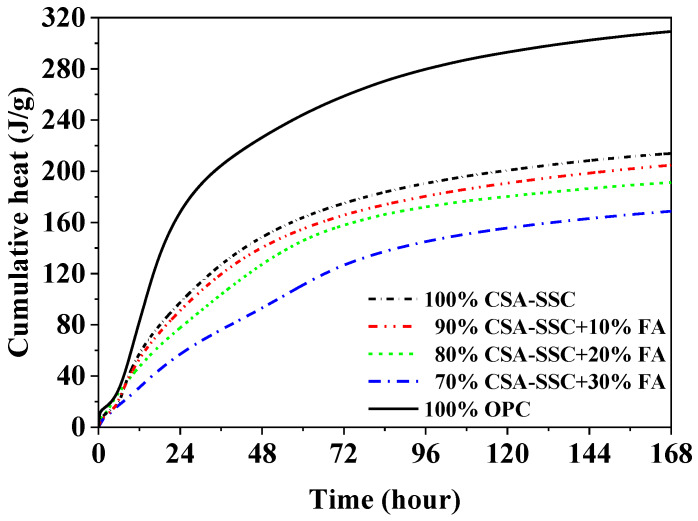
Cumulative hydration heat of CSA-SSC, CSA-SSC with FA, and OPC pastes.

**Figure 5 materials-13-02514-f005:**
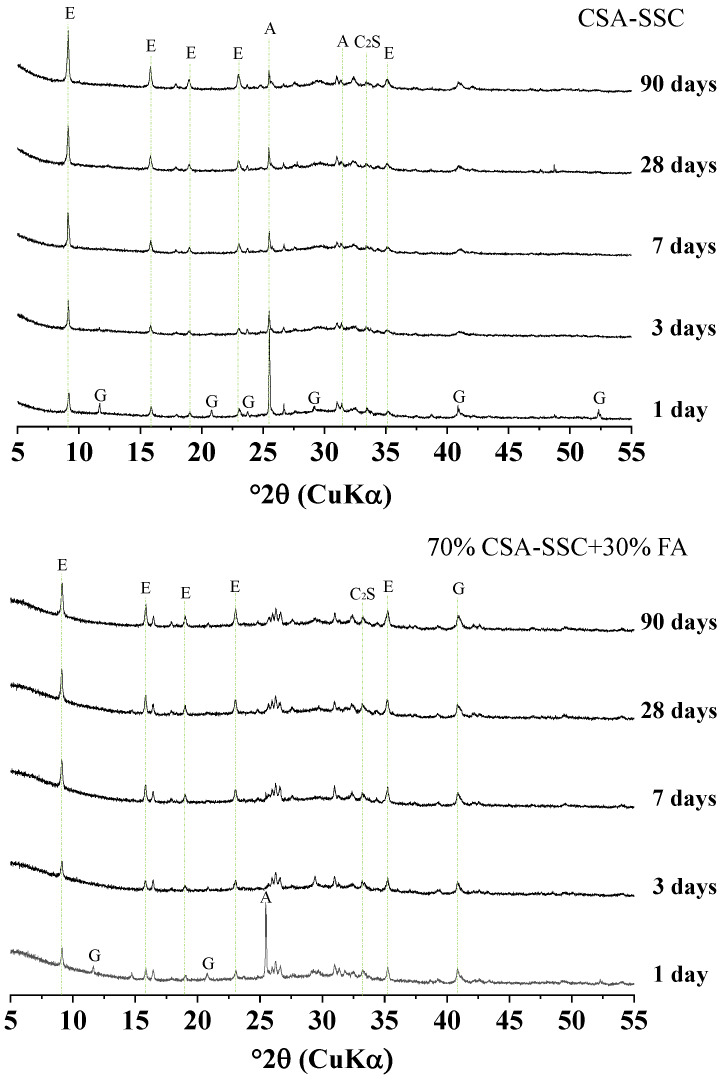
XRD analysis of the CSA-SSC and 70% CSA-SSC + 30% FA. (E: Ettringite, G: Gypsum, A: Anhydrite, C_2_S: Dicalcium silicate).

**Figure 6 materials-13-02514-f006:**
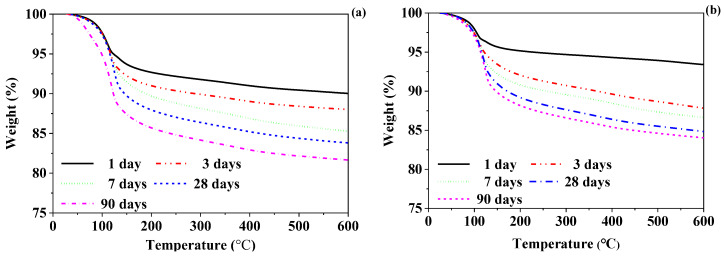
Thermogravimetric analysis (TGA) curves of (**a**) CSA-SSC and (**b**) 70% CSA-SSC + 30% FA pastes.

**Figure 7 materials-13-02514-f007:**
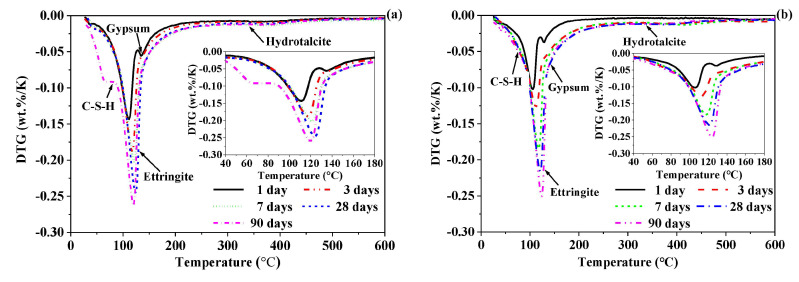
Differential thermogravimetry (DTG) curves of (**a**) CSA-SSC and (**b**) 70% CSA-SSC + 30% FA pastes.

**Figure 8 materials-13-02514-f008:**
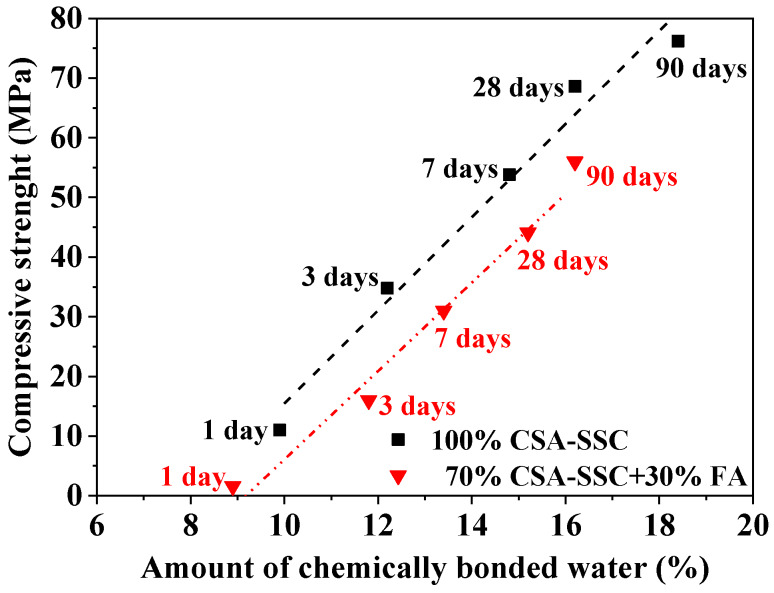
Relationship between compressive strength and amount of chemically bonded water of CSA-SSC and 70% CSA-SSC + 30% FA.

**Table 1 materials-13-02514-t001:** Chemical composition of calcium sulfoaluminate-activated supersulfated cement (CSA-SSC), fly ash (FA), and ordinary Portland cement (OPC).

Oxides/%	CaO	SiO_2_	Al_2_O_3_	MgO	SO_3_	Fe_2_O_3_	TiO_2_	LOI	Total	Blaine Values (cm^2^/g)
CSA-SSC	39.25	26.95	13.21	8.51	7.84	0.41	0.53	0.31	97.01	4500
FA	3.05	48.63	37.37	1.6	0.84	3.78	-	3.61	98.88	4200
OPC	64.31	19.79	5.31	2.52	2.47	2.88	0.34	1.46	99.08	3500

**Table 2 materials-13-02514-t002:** Mineralogical composition of the high-belite calcium sulfoaluminate cement (HB-CSA) clinker.

Mineralogical Composition/%	C_4_A_3_ $\bar{S}$	C_2_S	f-CaO	CaSO_4_	Other
HB-CSA clinker	28	45	4.6	16	6.4

**Table 3 materials-13-02514-t003:** Mixture proportions of the mortars. (Binders: cement and fly ash).

Sample	Water/Binders (wt./wt.)	Cement (wt.%)	Fly Ash (wt.%)	Binders/Sand (wt./wt.)
CSA-SSC	1:2	100	-	1:3
70% CSA-SSC + 30% FA	1:2	70	30	1:3
OPC	1:2	100	-	1:3

**Table 4 materials-13-02514-t004:** Mixture proportions of the pastes. (Binders: cement and fly ash).

Sample	Water/Binder (wt./wt.)	Cement (wt./%)	Fly Ash (wt./%)
CSA-SSC	2:5	100	-
70% CSA-SSC + 30% FA	2:5	70	30

**Table 5 materials-13-02514-t005:** Representative values of thermogravimetric analysis (TGA) and differential thermogravimetry (DTG) curves of hydration products adapted from [30].

Hydration Products	Temperature of the Major Peak	Temperature Range of Weight Loss
Ettringite	120 °C	50–400 °C
C-S-H	150 °C	50–600 °C
Gypsum	140 °C	100–170 °C
OH-hydrotalcite	250 °C and 400 °C	50–500 °C

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
