# Peer review of "Influence of Fly Ash on Mechanical Properties and Hydration of Calcium Sulfoaluminate-Activated Supersulfated Cement"

_materials, 2020, doi:10.3390/ma13112514_

Round 1

Reviewer 1 Report

  1. Abstract: “This paper aims to investigate” – the paper does not aim to investigate – In this paper, you are reporting the outcomes of your investigation. Maybe you should write “this paper aims to report…”?
  2. Table 1 Is not within the margins – partly presented in the next page.
  3. Provide Blaine values for table one Items
  4. Line 92 is there one European standard for by compressive and flexural strength
  5. Line 93 provide full details of the mixes including any additives not just the ratios - also discuss the method of preparation of the samples
  6. Provide full details of the specimens tested and their numbers also make a statement of how these samples were cured until the time of testing – This is very confusing -the details of specimens are provided in section  2.3
  7. line 99 Provide full details off there thermal metric TAM a machine
  8. Water to cement ratio for the samples what's 2.5 in line 105 is this a different mix to that present it in section 2.2.1?
  9. Line 105 prism dimensions are a repeat of those in line 93
  10. Remove the lines in the results of figures 1& 2
  11. Line 138 why are you referring to compressive and flexural strength at the same time? you should discuss 1 first and then the other - do not combine them - they're not the same thing.
  12. Place the error bars for each of the points or provide standard deviation
  13. Line 142 at 28 d of hydration - you mean after 28 days curing after samples? if not provide hydration test data first (section 3.2 (fix this error elsewhere in the paper) –
  14. Fgs 3&4 indicate a higher degree of hydration and reactivity of the ordinary Portland cement - when compared to 100% CSA -SSC mix; Accordingly, one would expect that 100% OPC mix have higher strength - That is not the case from figure 1 & 2 even at early age 2 days. – Do you have any explanation for this?
  15. Line 181 what does “tend to gentle” mean?

Author Response

Reviewer #1:

1

Comment

Abstract: “This paper aims to investigate” – the paper does not aim to investigate – In this paper, you are reporting the outcomes of your investigation. Maybe you should write “this paper aims to report…”?

Reply

Thank you very much for your careful review. We have modified it in the line 10 of revised paper.

(This paper aims to report the effects of fly ash (FA) on the mechanical properties and hydration of calcium sulfoaluminate-activated supersulfated cement (CSA-SSC).)

2

Comment

Table 1 Is not within the margins – partly presented in the next page.

Reply

Thank you very much for your careful review. We have modified it in the line 95 of revised paper.

3

Comment

Provide Blaine values for table one Items

Reply

Thank you very much for your careful review. We have modified it in the line 95 of revised paper.

4

Comment

Line 92 is there one European standard for by compressive and flexural strength

Reply

Thank you very much for your careful review. We have modified it in the line 101 of revised paper. The compressive and flexural strength of mortars are measured according to the European Standard DIN EN 196-1.

5

Comment

Line 93 provide full details of the mixes including any additives not just the ratios - also discuss the method of preparation of the samples.

Reply

Thank you very much for your careful review. We give a detailed table of the mortar compositions and modify the method of preparation of the samples in the line 99-109 and 132-133 of revised paper. In addition, the European Standard DIN EN 196-1 provide full details experimental methods, and we refer directly to the experiment.

6

Comment

Provide full details of the specimens tested and their numbers also make a statement of how these samples were cured until the time of testing – This is very confusing -the details of specimens are provided in section 2.3

Reply

Thank you very much for your careful review. We have modified it in the line 118-128 of revised paper.

7

Comment

line 99 Provide full details off there thermal metric TAM a machine

Reply

Thank you very much for your careful review. This thermal metric TAM is “Thermometric TAM Air Isothermal Calorimeter” in the line 113 of revised paper.

8

Comment

Water to cement ratio for the samples what's 2.5 in line 105 is this a different mix to that present it in section 2.2.1?

Reply

Thank you very much for your careful review. In mechanical properties test the water to binders (cement and fly ash) ratio for the mortars is 1:2, but the water to cement ratio for the pastes is 2:5.

9

Comment

Line 105 prism dimensions are a repeat of those in line 93

Reply

Thank you very much for your careful review. The mortars and pastes use the same prisms (40 mm × 40 mm × 160 mm)

10

Comment

Remove the lines in the results of figures 1& 2

Reply

Thank you very much for your careful review. The lines in the results of Figures 1 and 2 are used to better see the changing trend of the strength.

11

Comment

Line 138 why are you referring to compressive and flexural strength at the same time? you should discuss 1 first and then the other - do not combine them - they're not the same thing.

Reply

Thank you very much for your careful review. We have separately described the contents of compressive strength and flexural strength. We have modified it in the line 155 and 175 of revised paper.

12

Comment

Place the error bars for each of the points or provide standard deviation

Reply

Thank you very much for your careful review. We have modified it in the line 176 and 178 of revised paper.

13

Comment

Line 142 at 28 d of hydration - you mean after 28 days curing after samples? if not provide hydration test data first (section 3.2 (fix this error elsewhere in the paper)

Reply

Thank you very much for your careful review. “At 28 d of hydration” mean after 28 days hydration. We have modified it in the section 3.2 and all error elsewhere of revised paper.

14

Comment

Fig 3&4 indicate a higher degree of hydration and reactivity of the ordinary Portland cement - when compared to 100% CSA -SSC mix; Accordingly, one would expect that 100% OPC mix have higher strength - That is not the case from figure 1 & 2 even at early age 2 days. – Do you have any explanation for this?

Reply

Thank you very much for your careful review. The hydration mechanism of OPC is different from that of CSA-SSC, and the main hydration products of OPC are C-S-H and calcium hydroxide, but the main hydration products of CSA-SSC are ettringite and C-S-H. Although OPC has a high degree of early hydration and high early strength, at the early stage of CSA-SSC hydration, the hydration of C4A3 with CaSO4 leads to a rapid formation of the ettringite, and then the early strength of CSA-SSC will continue to increase. Ettringite has the effect of strengthening agent, and the strength CSA-SSC will rapidly enhance and exceed that of OPC after 2 days of hydration. Meanwhile, the hydration of f-CaO and C2S provides an alkaline environment for the activation of GBFS, which also produces ettringite. In the late stage, in addition to the continuous formation of ettringite, the GBFS continues to hydrate in an alkaline environment and forms C-S-H, leading to the increase in the late strength.

15

Comment

Line 181 what does “tend to gentle” mean?

Reply

Thank you very much for your careful review. The “tend to gentle” mean tend to be horizontal.

Reviewer 2 Report

The experimental campaign is well described. The conclusions are supported by the results.

I recommend the publication after the following minor modifications:

  1. L30 "causing cracking of the concrete", cracking appears only if the thermal deformations are restrained
  2. L32 “key to prevent the development of cracks " in massive concrete structures.
  3. L35-36 is ambiguous, please rewrite. I believe that the authors mean that SSC has low hydration heat and not clinker.
  4. L59 Please provide a reference for “The late strength of SSC mainly depends on the amount of C-S-H produced. ”
  5. Section 3.1, Fig 1. and 2. Please provide the standard deviations
  6. Is there a reason why the (main) peak of 80% CSA-SC system s shifted to the left when compared to the other systems? Maybe the t=0 was slightly different?
  7. L190 “main crystal hydration product" : main crystalline hydration product ?
  8. Fig 5. The peaks corresponding to ye’elimite are not identified. How is was possible to determine that “After 1 d of hydration, the  C4A3S in the CSA-SSC and 70% CSA-SSC+30% FA paste is not observed.” (L191-192)?
  9. L276-277: maybe suggest that a quantification of porosity in the system could provide an argument to corroborate this hypothesis?
  10. L286-287 “it is” instead of “it’s”
  11. A sentence on the discussion on the relationship between “strength and amount of chemically bonded water” could be relevant to the conclusion.

Author Response

Reviewer #2:

1

Comment

L30 "causing cracking of the concrete", cracking appears only if the thermal deformations are restrained.

Reply

Thank you very much for your careful review. We have modified it in the line 30-31 of revised paper.

2

Comment

L32 “key to prevent the development of cracks " in massive concrete structures.

Reply

Thank you very much for your careful review. We have modified it in the line 32-33 of revised paper.

3

Comment

L35-36 is ambiguous, please rewrite. I believe that the authors mean that SSC has low hydration heat and not clinker.

Reply

Thank you very much for your careful review. We have modified it in the line 37-40 of revised paper.

4

Comment

L59 Please provide a reference for “The late strength of SSC mainly depends on the amount of C-S-H produced.”

Reply

Thank you very much for your careful review. I have provided a reference in the line 63 of revised paper.

5

Comment

Section 3.1, Fig 1. and 2. Please provide the standard deviations

Reply

Thank you very much for your careful review. I have provided the standard deviations in the line 177 and 179 of revised paper.

6

Comment

Is there a reason why the (main) peak of 80% CSA-SC systems shifted to the left when compared to the other systems? Maybe the t=0 was slightly different?

Reply

Thank you very much for your careful review. The reason why the (main) peak of the 90% CSA-SC systems, 80% CSA-SC systems and 70% CSA-SC systems shifted to the left is no accurate explanation at present. The reason for this phenomenon may be due to the water-reducing effect of glassy FA.

7

Comment

L190 “main crystal hydration product": main crystalline hydration product?

Reply

Thank you very much for your careful review. We have modified it in the line 217-218 of revised paper.

8

Comment

Fig 5. The peaks corresponding to ye’elimite are not identified. How is was possible to determine that “After 1 d of hydration, the C4A3S in the CSA-SSC and 70% CSA-SSC+30% FA paste is not observed.” (L191-192)?

Reply

Thank you very much for your careful review. Because the content of the ye’elimite in the CSA-SSC and 70% CSA-SSC+30% FA paste is very small. Moreover, the ye’elimite reacts very quickly with anhydrite, and it has been completely reacted in a few hours. It can't be detected after 1 d of hydration.

9

Comment

L276-277: maybe suggest that a quantification of porosity in the system could provide an argument to corroborate this hypothesis?

Reply

Thank you very much for your careful review. We are very sorry that we did not do a quantitative experiment of the porosity. Because this paper mainly studies the effect of fly ash on mechanical properties and hydration of calcium sulfoaluminate-activated supersulfated cement. However, your comments are very helpful to us, and we will add this part in the future research.

10

Comment

L286-287 “it is” instead of “it’s”

Reply

Thank you very much for your careful review. We have modified it in the line 3-7 of revised paper.

11

Comment

A sentence on the discussion on the relationship between “strength and amount of chemically bonded water” could be relevant to the conclusion.

Reply

Thank you very much for your careful review. We have modified it in the line 321-322 of revised paper.

(The compressive strength of the CSA-SSC and 70% CSA-SSC+30% FA has a linear growth relationship with the amount of chemically bound water.)

Reviewer 3 Report

Dear editor,

The authors analyze the partial substitution of fly ash (FA) in calcium sulfoaluminate-activated supersulfated cement (CSA-SSC). They show experimental results in which they compare the mechanical properties and hydration of various mixtures (FA) with (CSA-SSC) in relation to pure cement (CSA-SSC) and Ordinary Portland cement (OPC) commonly used in construction.

I provide a frank description of the article's strengths and weaknesses: 

Strengths:

This work presents a good scientific level, with an interesting study and solid conclusions.

Appropriate scientific instrumentation has been used to determine the properties studied.

Weaknesses:

Despite the good scientific level, it would be interesting in the discussion or conclusions to refer to the limitations of the study.

The introduction justifies the use of cements with low heat and a low environmental impact. It would be interesting to make some indication about those fields in which this cement is appropriate and the limitations it may have, as well as its possible behavior against other unanalyzed characteristics such as durability of steel reinforced concrete

Based on the strengths and weaknesses of the manuscript, I recommend menor revision of the paper.

Specific comments on the paper:

in line 42 the authors affirm “The SSC requires less energy and produces lower greenhouse gas emissions than the OPC”

What is this statement based on? The authors should include an appropriate reference, with data on the energy reduction and CO2 emissions to support it. 

In line 43 the authors affirm “in addition to the environmental advantages, the SSC exhibits mechanical behaviors, such as high durability in sulfate solutions, low hydration heat, and excellent resistance to chloride, acid, and seawater attacks, which all confirm than the SSC is an alternative to Portland Cement”

Please, review this sentence. Did the authors want to refer to chemical or thermal behaviours, instead of mechanical?

In line 73 the authors affirm “In this study, the mechanical properties of CSA-SSC mixed with FA are compared and analyzed with the strength of reference mortar prepared by OPC”

Please, review English. Additionally, maybe the authors should also indicate that properties of the CSA-SSC system (without FA) are also compared. 

In line 86 the authors affirm “The specific surface areas of CSA-SSC, FA and OPC were 4500cm2/g, 4200cm2/g and 3500 cm2/g, respectively.

Given that these data have been provided, together with reference 8, which deals with the influence of the fineness of the powder on the SCC hydration, it would be interesting that the authors completed their study with an analysis on how this parameter influences the hydration of the different systems analyzed. Thus, the authors could determine how the setting time of the different mixtures developed varies, which can be determined according to EN 196-3 Methods of testing cement. Determination of setting times and soundness.

Author Response

Reviewer #3:

1

Comment

in line 42 the authors affirm “The SSC requires less energy and produces lower greenhouse gas emissions than the OPC”

What is this statement based on? The authors should include an appropriate reference, with data on the energy reduction and CO2 emissions to support it.

Reply

Thank you very much for your careful review. The SSC is comprised of 80−85% granulated blastfurnace slag (GBFS) (industrial by-products), 10−15% gypsum or anhydrite, and < 5% ordinary Portland cement (OPC) clinker or lime as the alkali activator. In addition, anhydrite or gypsum can also be replaced by other industrial by-products such as phosphogypsum and fluorgypsum. Other industrial by-products and waste can also be added to the SSC system. Therefore, we believe that SSC is less energy in the production process, and uses a lot of industrial waste and by-products. We have added references in the line 46 of revised paper.

2

Comment

In line 43 the authors affirm “in addition to the environmental advantages, the SSC exhibits mechanical behaviors, such as high durability in sulfate solutions, low hydration heat, and excellent resistance to chloride, acid, and seawater attacks, which all confirm than the SSC is an alternative to Portland Cement” 

Please, review this sentence. Did the authors want to refer to chemical or thermal behaviours, instead of mechanical?

Reply

Thank you very much for your careful review. We have modified it in the line 47 of revised paper.

(the SSC exhibits chemical or thermal behaviors, such as high durability in sulfate solutions, low hydration heat, and excellent resistance to chloride, acid, and seawater attacks, which all confirm that the SSC is an alternative to Portland cement.)

3

Comment

In line 73 the authors affirm “In this study, the mechanical properties of CSA-SSC mixed with FA are compared and analyzed with the strength of reference mortar prepared by OPC”

Please, review English. Additionally, maybe the authors should also indicate that properties of the CSA-SSC system (without FA) are also compared.

Reply

Thank you very much for your careful review. We have modified it in the line 162-165 of revised paper.

(The CSA-SSC exhibits higher the compressive and flexural strengths than those of 90% CSA-SSC+10% FA, 80% CSA-SSC+20% FA and 70% CSA-SSC+30% FA at all hydration ages. The CSA-SSC exhibits higher late compressive and flexural strengths and a larger increase in the compressive strength than the OPC.)

4

Comment

In line 86 the authors affirm “The specific surface areas of CSA-SSC, FA and OPC were 4500cm2/g, 4200cm2/g and 3500 cm2/g, respectively.

Given that these data have been provided, together with reference 8, which deals with the influence of the fineness of the powder on the SCC hydration, it would be interesting that the authors completed their study with an analysis on how this parameter influences the hydration of the different systems analyzed. Thus, the authors could determine how the setting time of the different mixtures developed varies, which can be determined according to EN 196-3 Methods of testing cement. Determination of setting times and soundness.

Reply

Thank you very much for your careful review. We are very sorry that we haven't done setting times and soundness. Because this paper mainly studies the effect of fly ash on mechanical properties and hydration of calcium sulfoaluminate-activated supersulfated cement. However, your comments are very helpful to us, and we will do this part in the future research.

Reviewer 4 Report

Recommendations
No photos of the analyzed raw materials. Complete the morphology results, e.g. SEM images of the surface after the compressive strength test.

Author Response

Reviewer #4:

1

Comment

No photos of the analyzed raw materials. Complete the morphology results, e.g. SEM images of the surface after the compressive strength test.

Reply

Thank you very much for your careful review. We are very sorry that we did not test the SEM images of the surface after the compressive strength test. Because this paper mainly studies the effect of fly ash on mechanical properties and hydration of calcium sulfoaluminate-activated supersulfated cement. However, your comments are very helpful to us, and we will add this part in the future research.

Reviewer 5 Report

the article is well written in part, it is neat. the figures are clear.

I have some questions:

what is the level of Fa in mortars? is it addition or substitution to cement?

it would be interesting to put a detailed table of the mortar compositions (ciment or FA and sand and water) with the Water/Ciment or Water/Binder ratios. And has the amount of water been recalculated with the FA addition ?

what does ICC mean? personally, I prefer full titles for the parts (example: XRD = X-Ray Diffractometry).

Have you studied the pozzolinic activity of FA (standard ASTM C618)? it would be interesting to add a paragraph and to calculate the activity index compared to OPC cement.

Do the results in bending bring complementary conclusions to those in compression? what is the point of keeping the results in flexion?

Author Response

Reviewer #5:

1

Comment

what is the level of Fa in mortars? is it addition or substitution to cement?

Reply

Thank you very much for your careful review. The types of fly ash (FA) in this paper is Class F. FA is a substitution to cement.

2

Comment

it would be interesting to put a detailed table of the mortar compositions (cement or FA and sand and water) with the Water/Cement or Water/Binder ratios. And has the amount of water been recalculated with the FA addition?

Reply

Thank you very much for your careful review. We have modified it in the line 108-109 and 132-133 of revised paper. FA is considered to be binders, so there is no need to recalculate the amount of water.

3

Comment

what does ICC mean? personally, I prefer full titles for the parts (example: XRD = X-Ray Diffractometry).

Reply

Thank you very much for your careful review. We have modified it in the line 110, 134, 141 of revised paper.

4

Comment

Have you studied the pozzolanic activity of FA (standard ASTM C618)? it would be interesting to add a paragraph and to calculate the activity index compared to OPC cement.

Reply

Thank you very much for your careful review. We are very sorry that we haven't studied the pozzolanic activity of FA yet. Because this paper mainly studies the effect of fly ash on mechanical properties and hydration of calcium sulfoaluminate-activated supersulfated cement. However, your comments are very helpful to us, and we will add this part in the future research.

5

Comment

Do the results in bending bring complementary conclusions to those in compression? what is the point of keeping the results in flexion?

Reply

Thank you very much for your careful review. Compressive strength and flexural strength are important standards for mechanical properties of cementitious materials. In addition, through testing the compressive strength and flexural strength of cementitious materials, we can know the application performance of cementitious materials.

Round 2

Reviewer 2 Report

The authors have revised the article taking into account my comments. I recommend publication in the present form.

Author Response

Thank you for your comments concerning our manuscript entitled “Influence of fly ash on mechanical properties and hydration of calcium sulfoaluminate-activated supersulfated cement” (ID: materials-806646) .Those comments are all valuable and very helpful for revising and improving our paper, as well as the important guiding significance to our researches.

Reviewer 3 Report

Dear Authors,

Thank you for your corrections in the manuscript entitled “Influence of fly ash on mechanical properties and hydration of calcium sulfoaluminate-activated supersulfated cement” (ID: materials-806646) about our comments.

Comments and corrections made in relation to our review have been carefully studied.

You have attended most of the comments made in the first review.

A document is attached, with the evaluations to your answers.

Best wishes

Author Response

Dear Reviewers:

   Thank you for your comments concerning our manuscript entitled “Influence of fly ash on mechanical properties and hydration of calcium sulfoaluminate-activated supersulfated cement” (ID: materials-806646) .Those comments are all valuable and very helpful for revising and improving our paper, as well as the important guiding significance to our researches. We have studied comments carefully and have made correction which we hope meet with approval. According to expert opinion, revised portion are marked in red in the paper. The main corrections in the paper and the responds to the reviewer’s comments are as following:

Reviewer #3:

3

Comment

In line 73 the authors affirm “In this study, the mechanical properties of CSA-SSC mixed with FA are compared and analyzed with the strength of reference mortar prepared by OPC”

Please, review English. Additionally, maybe the authors should also indicate that properties of the CSA-SSC system (without FA) are also compared.

Reply

Thank you very much for your careful review. We have modified it in the line 76 of revised paper.

(In this study, the mechanical properties of CSA-SSC mixed with FA are compared and analyzed with the strength of reference mortar prepared by CSA-SSC without FA and OPC.)

4

Comment

In line 86 the authors affirm “The specific surface areas of CSA-SSC, FA and OPC were 4500cm2/g, 4200cm2/g and 3500 cm2/g, respectively.

Given that these data have been provided, together with reference 8, which deals with the influence of the fineness of the powder on the SCC hydration, it would be interesting that the authors completed their study with an analysis on how this parameter influences the hydration of the different systems analyzed. Thus, the authors could determine how the setting time of the different mixtures developed varies, which can be determined according to EN 196-3 Methods of testing cement. Determination of setting times and soundness.

Reply

Thank you very much for your careful review. The study of setting time and soundness has not been done. Because our research focuses on the influence of fly ash on mechanical properties and hydration of CSA-SSC, the part of experiments on setting time and soundness has little to do with our research objectives, so I did not do it. However, your comments are very helpful to us, and the study of setting time and soundness will be carried out in the future.

We tried to our best to improve the manuscript and made some changes in the manuscript. These changes will not influence the content and framework of the paper. And here we did not list the changes but marked in yellow in revised paper.

Once again, thank you very much for your comments and suggestions.

Reviewer 4 Report

I accept the authors' reply. I recommend publishing this article.

Author Response

Dear Reviewers:

   Thank you for your comments concerning our manuscript entitled “Influence of fly ash on mechanical properties and hydration of calcium sulfoaluminate-activated supersulfated cement” (ID: materials-806646) .Those comments are all valuable and very helpful for revising and improving our paper, as well as the important guiding significance to our researches.

Reviewer 5 Report

The corrections made are accepted. I agree to the publication.

Author Response

(The authors gave the same response as above.)
